# Financial inclusion and physical health functioning among aging adults in the sub-Saharan African context: Exploring social networks and gender roles

**Razak M. Gyasi**[1]*, **Siaw Frimpong**[2], **Gilbert Kwabena Amoako**[3], **Anokye M. Adam**[2]

**1** African Population and Health Research Center (APHRC), Nairobi, Kenya, **2** Department of Finance, School of Business, University of Cape Coast, Cape Coast, Ghana, **3** Department of Accountancy and Accounting Information Systems, Kumasi Technical University, Kumasi, Ghana

* RGyasi@aphrc.org, Razak.MGyasi@gmail.com

## Abstract

### Background

It remains poorly understood how financial inclusion influences physical health functioning in later life in sub-Saharan African context and whether the association differs by gender and social relationships. We aim 1) to examine the associations of financial inclusion with functional impairment during older age in Ghana; and 2) to evaluate whether gender and social networks modify this association.

### Methods

The cross-sectional analyses are based on a sample ($N$ = 1,201) of study participants aged 50 years and over ($M$ = 66.5 years, $SD$ = 11.9, 63.3% female) deriving from the 2016–2017 AgeHeaPsyWel-HeaSeeB Study. Ordinary least squares (OLS) regression analyses with interactions were performed to estimate the link between financial inclusion and functional health and how the association is modified by gender and older age social networks.

### Results

The mean financial inclusion score was 1.66 (SD = 1.74) in women and 2.33 (SD = 1.82) in men whilst mean activities of daily living (ADL) score was 13.03 (SD = 4.99) and 14.85 (SD = 5.06) in women and men respectively. We found that financial inclusion was associated with decreases in ADL (total sample: $\beta$ = -.548, $p$ < .001; women: $\beta$ = -.582, $p$ < .001; men: $\beta$ = -1.082 $p$ < .001) and instrumental ADL (IADL) (total sample: $\beta$ = -.359, $p$ = .034; women: $\beta$ = -.445, $p$ = .026 but not in men). Social networks significantly moderated the association of financial inclusion with ADL such that the financially included who were embedded in a stronger constellation of social networks were 6% less likely to report ADL impairment compared to those with weaker social networks ($\beta$ = -.062, $p$ = .025).

**Data Availability Statement:** All relevant data are within the paper and its Supporting Information files and are freely available for public access and use without any institutional restrictions.

**Funding:** This work was supported by Lingnan University, Hong Kong [RPG 1129310] to RMG (https://www.ln.edu.hk/about-lu/introducinglingnan). The funders had no role in study design, data collection and analysis, decision to publish, or preparation of the manuscript.

**Competing interests:** The authors have declared that no competing interests exist.

## Conclusions

The study provides empirical evidence for a better understanding of the association between financial inclusion and physical health functioning in the context of later life social networks. Interventions for functional health through financial inclusion in sub-Saharan Africa should include improving interpersonal and social networks for older adult and also through gender lenses.

## Introduction

Trends in demographic aging in sub-Saharan Africa have been massive with serious implications for family structure, social support and public health including functional status [1]. Research has shown that functional capabilities are negatively related to age, whilst the ability to participate in financial market correlates with daily activity limitations [2–4]. Interventions targeted at promoting financial inclusion in later life can be innovative in contributing to functional health policy and healthy aging framework in low- and middle-income settings [5–7]. Despite this broad knowledge base, limited research has evaluated the proximate roles of social factors about the effect of financial inclusion on functional health in sub-Saharan Africa.

Financial inclusion seeks to increase access, safety and use of financial services such as bank and money transfer accounts to enhance welfare and mitigate shocks among the poor [7–9]. Easy access to financial services among aging adults who have irregular incomes, largely due to retirement, and social isolation has been identified as a major social determinant of health in later life [2]. Crucially, financial inclusion is recognized as a mechanism to boost health and wellbeing and also as an enabler for the universal health coverage, outlined in the Sustainable Development Goal 3 [6–10]. The United Nations Secretary-General's Special Advocate for Inclusive Finance for Development [11] has demonstrated a positive relationship between universal access to financial services and improvement in health. Financial inclusion programs may offer broader opportunities than traditional income proxies and are crucial for older people who are likely to live with functional impairment, physical inactivity and often with financial vulnerabilities [3–6]. However, the Findex data show that approximately 40% of adults globally are not only unbanked but also almost alienated from basic financial activities especially in developing countries [12].

Financial capability provides opportunities for older people to take greater control of their finances, external environments, and be able to manage economic resources better and to achieve better health outcomes [2, 13]. Research, particularly in advanced countries shows that financial literacy and the attendant inclusion are negatively associated with common mental health problems such as depression, psychological distress, anxiety, and poor overall health outcomes [2, 14, 15]. Older people who maintain financial aptitude through access to a bank account report better overall health [2, 16, 17]. Among older Hispanics in the United States, Aguila et al. [15] observed that bank account ownership was associated with improved mental health. Kim et al. [18] analyzed data from 176 countries and found that gender inequalities in bank account access were associated with higher female to male stroke ratio. Similarly, in the United States, Finkelstein et al. [19] observed that access to financial services such as health insurance for the uninsured improves the wellbeing by providing financial protection, reducing stress, and ultimately, enhanced both mental and physical health outcomes of older people who may lack regular incomes due to retirement and other socioeconomic circumstances. Improving older adults' ability to manage their finances remains a fundamental step to promoting financial capabilities, and in turn, improve their wellbeing and overall health [13, 20].

The financial capability provides practical implications for public health of various demographic cohorts by maintaining reasonable financial alertness and reducing health risks [21–23]. Although being financially capable may relate to a wide range of socioeconomic factors, it has a greater influence on mental and physical health [23]. Using the British Household Panel Survey data from 1991–2006, Atkinson [21] documents a positive relationship between financial capability and psychological wellbeing among adults. Findings from studies in various low-income countries have reported the value of financial inclusion measures such as bank, Susu and mobile money accounts ownership as a means to support older people and women's health and services use behavior [2, 18, 24–26].

Despite several attempts to investigate the impact of financial inclusion on health, the present state of evidence leaves it insufficiently clear to understand the dynamics of functional inclusion particularly for the ever-growing older population in low-income settings [2]. The only study we are aware of did not establish an association between physical health and ownership of a bank account across socioeconomic groups among the Hispanics [15]. The present study examines the overall and gender-wise associations between financial inclusion and functional impairment among older people and to determine whether the associations are differentiated by social networks. Exploring these relationships is crucial because functional health in later life is noted to differ between genders [27, 28].

Studies also demonstrate that women have less access to formal financial services or less financially included than the average population [7, 29]. Similarly, research indicates that social resources provide the requisite financial education and the related financial literacy may contribute to good financial decision making [10, 29–31]. Thus, improving financial literacy through social networks is useful for improving financial inclusion. Identifying the role of social networks could proffer valuable insights into how financial inclusion influences functional impairment. Yet, there is a few published literature on the role of social networks in relation to the effect of financial inclusion on functional status of older persons in sub-Saharan Africa. We, therefore, hypothesized that older adults who are financially included are less likely to report functional impairment. It is further expected that the associations of financial inclusion with functional impairment are modified by social networks and gender sub-groups.

## Methods

### Design, sample and data

This paper included a sample of older people from the Aging, Health, Wellbeing and Health-seeking Behavior Study (AgeHeaPsyWel-HeaSeeB Study), a representative cross-sectional study in Ghana [32]. Conducted in 2016–2017, the AgeHeaPsyWel-HeaSeeB Study, considered men and women aged 50 years and over from nine urban and 15 rural communities in six districts of Ashanti Region, who participated in the survey taking into consideration geographical variation and to improve representation. The survey was based on a stratified random sample of households in the communities. The sample size was determined using the formula, $n = design\ effect \times [(Z_{\alpha/2})^2 \times P(1-P)]/\varepsilon^2$ [33] with 5% margin of error, 95% confidence interval, design effect of 1.5, type 1 error of 5%, type 2 error of 15%, and a conservative estimation of 50%. The statistical power calculation revealed that the sample size had 85% power to detect an odds ratio of $\geq 2$. A total of 1,200 eligible participants were interviewed with a response rate 98.4%. Details on the selection procedure and response rate are provided elsewhere [2, 34, 35]. The data were collected via in-person structured interviews conducted by trained Asante Twi-speaking research assistants with research background in public health and social policy.

The study received ethics approval from the Committee on Human Research Publication and Ethics, School of Medical Sciences, Kwame Nkrumah University of Science and Technology and Komfo Anokye Teaching Hospital, Kumasi, Ghana (Ref: CHRPE/AP/507/16) as well as the Research Ethics Committee of Lingnan University, Hong Kong. Informed written and oral consent was obtained from all participants.

## Measures

**Functional health status.**   The outcome variable of interest was functional impairment operationalized based on the activities of daily living (ADL) and instrumental ADL (IADL). Self-reported difficulty in performing these activities in the last 30 days with good psychometric properties [36] was assessed. ADL were measured using a six-item scale, which assesses the ability or difficulty in conducting the following activities on their own: eating, bathing or washing the whole body, dressing up or putting on clothes, getting in or out of bed, and using the toilet and moving around inside your house. These responses were assessed on a four-point scale ranging from 1 = not limited at all, 2 = less limited, 3 = somewhat limited, and 4 = much limited. The total score ranging from 1 to 24 with a higher score indicating greater ADL impairment [37]. IADL were addressed using a seven-item scale relevant to the local conditions, including using the telephone/mobile phone, using public transport, shopping for groceries, preparing meals/kitchen chores, washing of clothes/doing laundry, medicine management and ability to manage finances [38]. We assessed the responses on a four-point scale ranging from: 1 = not limited at all, 2 = less limited, 3 = somewhat limited and 4 = much limited with a total score ranging from 1 to 28. A higher score indicated greater IADL impairment.

**Financial inclusion.**   The main explanatory variable was financial inclusion. Respondents provided a 1 = no or 2 = yes response to items on their involvement in basic financial services or instruments over the past 12 months. These included ownership of a personal bank account, withdrawal of money from an account, use of automatic teller machines, membership of a credit union, ownership of a 'Susu' account (system of informal savings scheme between a small group of people), easy access to loans from financial or non-financial institutions, ownership a Mobile Money Service account and having active National Health Insurance Scheme card. A possible total score ranged from 0 to 8 with higher scores suggesting higher levels of financial inclusion [2].

**Other variables.**   Several control variables that may confound the associations between financial inclusion and physical functioning were included in our regression model [2, 16]. These included age (categorized into 50–59, 60–69 and 70+ years), gender (male/female), residence (rural/urban), employment status, treated as a proxy for the financial situation (unemployed/employed), and living with spouse (yes/no). Level of educational reflected the highest obtained education and was categorized into primary school/no attendance, secondary education, and higher. Social networks was assessed with past 30-day contacts, involvement and social participation in the community through interaction with family, friends and neighbors and participation in social activities including attending religious services, social organizations, sports and cultural activities and civic or political clubs [39]. The responses for these items were recorded on a five-point scale: 1 = never, 2 = less frequently, 3 = frequently, 4 = very frequently, and 5 = every day. The overall score ranged from 4 to 20 with a higher scores indicating higher levels of social networks.

The health-related variables were included. Self-rated health was assessed with an item asking respondents to evaluate their general state of health with a 4-point Likert response scale: 1 = poor, 2 = fair, 3 = good, and 4 = good/excellent. Chronic conditions were assessed with a

question, "Did a doctor or health professional ever told you that you had. . .?" The list of conditions included hypertension, diabetes, respiratory diseases, cancers, stroke, chronic kidney diseases, asthma, arthritis, depression and insomnia. Mental distress was assessed with the Kessler Psychological Distress Scale (K-10) [40]. Each item had five response category: 1 = none of the time, 2 = a little of the time, 3 = some of the time, 4 = most of the time and 5 = all of the time. A sum score ranging from 10 to 50 was realized with higher scores reflecting higher levels of psychological distress.

## Statistical methods

The statistical analyses were conducted using SPSS v.21.0 (IBM, Armonk, NY) with $p < .05$ as statistically significant level. We first conducted univariate descriptive statistics for each variable of interest to contextualize the sample. This was followed by bivariate associations of the variables stratified by gender using Chi-square tests and Fishes exact tests for the categorical variables and independent $t$-test for the continuous variables. Multiple OLS regressions were separately conducted to predict the indicators of functional impairment, ADL and IADL limitations by the composite variable of financial inclusion, controlling for potential sociodemographic, health and support confounders. In the first step, analysis involving the overall sample was conducted. The second step estimated the gender-specific effects of financial inclusion on the outcomes. This was crucial because the financial inclusion and the magnitude of functional impairment in later life have been noted to differ by gender [2]. The final stage included the interaction terms in the pooled sample to investigate whether social networks potentially modify the association between financial inclusion and functional limitations. In a sensitivity analysis, we fitted a separate model to estimate the effect of each financial instrument on ADL and IADL outcome measures. Before the regression analysis, we conducted diagnostic tests to check for multicollinearity using the variance inflation factor (VIF). In our analysis none of the VIF scores exceeded the value of 1.31, suggesting no evidence for multicollinearity.

## Results

### Sample characteristics

Table 1 shows the descriptive statistics of the observations included in the OLS regression models stratified by gender. The average age was 66.5 years (SD: 11.9) and 63% were women. The socioeconomic status of the sample was quite low: the majority (86%) received primary or no education and more than half were not employed (56%). The mean social networks score was 6.10 (SD: 2.68) and about 62% of the participants were not living with their spouses. In terms of health status, about 49% rated their overall health as fair or poor and the majority (53%) reported living with at least one chronic condition. The average ADL, IADL and psychological distress scores were 13.70 (SD: 5.09), 6.32 (SD: 1.74) and 14.87 (SD: 6.21) respectively and the mean financial inclusion score was 1.91(SD: 1.79).

### Bivariate results

Approximately, 24% of men and 8% of women had achieved secondary education or higher-level education ($\chi^2$ = 58.545, $p < .001$) and the prevalence of unemployment among women (61%) was higher than in men (47%) ($\chi^2$ = 20.012, $p < .001$). Although women were less likely to be financially included (1.7 vs 2.3, $t$ = 35.619, $p < .039$) and lived with their spouses (74% vs 41%, $\chi^2$ = 133.332, $p < 0.001$), they were more likely to have higher or stronger levels of social networks compared to men (2 vs 1.8, $t$ = 41.741, $p = .041$). More women than men reported

**Table 1. Descriptive statistics and bivariate analysis of the sample.**

| Variables | Total | | Women (*n* = 759) | | Men (*n* = 441) | | *p*-value |
|---|---|---|---|---|---|---|---|
| | M or N | (SD or %) | M or N | (SD or %) | M or N | (SD or %) | |
| Age (in years, %) | | | | | | | |
| 50–59 | 426 | (35.5) | 272 | (35.8) | 154 | (34.9) | 0.478 |
| 60–69 | 343 | (28.6) | 208 | (27.4) | 135 | (30.6) | |
| 70+ | 431 | (35.9) | 279 | (36.8) | 152 | (34.5) | |
| Gender (women, %) | 759 | (63.3) | - | - | - | - | - |
| Residence (urban, %) | 660 | (55.0) | 430 | (56.7) | 230 | (52.2) | 0.131 |
| Educational level (%) | | | | | | | |
| None or Primary | 1034 | (86.2) | 697 | (91.8) | 337 | (76.4) | <0.001 |
| Secondary | 104 | (8.7) | 44 | (5.8) | 60 | (13.6) | |
| Tertiary | 62 | (5.2) | 18 | (2.4) | 44 | (10.0) | |
| Employment (not employed, %) | 667 | (55.6) | 459 | (60.5) | 208 | (47.2) | <0.001 |
| Living without spouse (%) | 742 | (61.8) | 563 | (74.2) | 179 | (40.6) | <0.001 |
| Social networks score | 6.10 | (2.68) | 2.01 | (1.43) | 1.82 | (1.48) | 0.041 |
| Self-rated health (%) | | | | | | | |
| Very good/excellent | 239 | (19.9) | 113 | (14.9) | 126 | (28.6) | <0.001 |
| Good | 369 | (30.8) | 234 | (30.8) | 135 | (30.6) | |
| Fair | 348 | (29.0) | 233 | (30.7) | 115 | (26.1) | |
| Poor | 244 | (20.3) | 179 | (23.6) | 65 | (14.7) | |
| Diagnosed chronic disease (yes, %) | 636 | (53.0) | 435 | (57.3) | 201 | (45.6) | <0.001 |
| Psychological distress score | 14.87 | (6.21) | | | | | |
| Financial inclusion score | 1.91 | (1.79) | 1.66 | (1.74) | 2.33 | (1.82) | 0.039 |
| Functional impairment | | | | | | | |
| IADL score | 6.32 | (1.74) | 6.13 | (1.76) | 6.65 | (1.66) | 0.087 |
| ADL score | 13.70 | (5.09) | 13.03 | (4.99) | 14.85 | (5.06) | <0.001 |

*Note*: M–mean; SD–standard deviation; N–number of participants; %–percentage.

poor or fair overall health (54% vs 41%, $\chi^2$ = 39.011, *p* < .001) and incidence of chronic condition (57% vs 46%, $\chi^2$ = 15.418, *p* < .001) but less likely to report ADL impairment (13 vs 15, *t* = 79.231, *p* < .005).

## Main regression analyses

In Tables 2 and 3, we presented the OLS regression models in which the ADL and IADL outcomes were regressed on the financial inclusion index. Model 1 presents the results of the overall sample, Models 2 and 3 show the gender-wise results for women and men respectively whilst Model 4 depicts the results of the interaction effects. All models were adjusted for the same set of socioeconomic, support and health-related control variables. In Table 2, the regression revealed that financial inclusion was significantly associated with a decrease in ADL impairment in the total sample (β = -.548, *p* < .001) in women (β = -.582, *p* < .001) and in men (β = -1.082 *p* < .001). The interaction between financial inclusion and social networks significantly moderated the association between financial inclusion and ADL impairment (β = -.062, *p* = .025) such that financially included individuals who were imbedded in stronger social networks reported lower ADL impairment than their counterparts with weaker social networks.

**Table 2. Multivariate OLS regressions predicting ADL impairment with composite score of financial inclusion and covariates.**

| | Model 1 –ADL: overall | | Model 2 –ADL: women | | Model 3 –ADL: men | | Model 4 –ADL: Interaction– financial inclusion × social networks | |
|---|---|---|---|---|---|---|---|---|
| | β | (SE) | β | (SE) | β | (SE) | β | (SE) |
| Financial inclusion (0–8; higher values indicating higher levels of financial inclusion) | -0.548*** | (0.154) | -0.582*** | (0.183) | -1.082*** | (0.323) | -0.544*** | (0.155) |
| Age (years, ref: 50–59) | 1 | | 1 | | 1 | | 1 | |
| 60–69 | 0.386* | (0.173) | 0.115*** | (0.219) | 0.716* | (0.307) | 0.387* | (0.173) |
| 70+ | 0.922*** | (0.175) | 0.888** | (0.220) | 0.895** | (0.322) | 0.923*** | (0.175) |
| Gender (ref: Males) | 1 | | | | | | | |
| Females | -0.341* | (0.153) | - | - | - | - | -0.343* | (0.153) |
| Residence (ref: rural) | | | | | | | | |
| Urban | -0.063 | (0.137) | -0.296 | (0.175) | 0.330 | (0.247) | -0.062 | (0.137) |
| Education (ref: Primary/ none) | 1 | | 1 | | 1 | | 1 | |
| Secondary | 0.659** | (0.246) | 1.535*** | (0.371) | -0.077 | (0.360) | 0.658** | (0.246) |
| Higher | 0.912** | (0.306) | 1.305* | (0.555) | 0.850* | (0.418) | 0.913** | (0.306) |
| Employment (ref: unemployed) | | | | | | | | |
| Employed | -0.689*** | (0.149) | -0.682*** | (0.191) | -1.031*** | (0.264) | -0.690*** | (0.150) |
| Living alone | -0.103 | (0.156) | 0.140 | (0.204) | -0.738** | (0.281) | -0.103 | (0.156) |
| Social networks score | -0.368** | (0.137) | -0.150 | (0.175) | -0.793*** | (0.245) | -0.379* | (0.147) |
| Self-reported health (ref: poor) | 1 | | 1 | | 1 | | 1 | |
| Fair | -1.765*** | (0.241) | -1.481*** | (0.305) | -2.546*** | (0.474) | -1.768*** | (0.241) |
| Good | -1.716*** | (0.214) | -1.697*** | (0.255) | -2.306*** | (0.455) | -1.717*** | (0.214) |
| Very good | -1.513*** | (0.211) | -1.558*** | (0.249) | -2.116*** | (0.468) | -1.515*** | (0.211) |
| Diagnosed chronic conditions | 0.692*** | (0.137) | .606*** | (0.174) | 0.970*** | (0.246) | 0.690*** | (0.138) |
| Psychological distress sore | 0.382** | (0.141) | 0.146 | (0.177) | 0.826*** | (0.259) | 0.383** | (0.141) |
| **Interaction effect:** financial inclusion × social networks | | | | | | | -0.062* | (0.289) |
| Constant | 0.556*** | (0.139) | 0.758*** | (.238) | 0.803*** | (0.229) | 0.555*** | (0.139) |
| Model diagnostics (-2 log likelihood) | 1348.692 | | 857.999 | | 446.211 | | 1348.646 | |
| Adjusted R² | 0.307 | | 0.301 | | 0.416 | | 0.307 | |

Beta-Coefficients are reported; Cluster-robust standard errors in parentheses.

Model 1: overall sample, Model 2: women; Model 3: men, Model 4: interaction analysis (financial inclusion × social network).

*** $p < 0.001$

** $p < 0.005$

* $p < 0.05$.

Similarly, our results in Table 3 showed that financial inclusion was negatively associated with IADL limitations in the pooled sample (β = -.359, $p$ = .034) and in women (β = -.445, $p$ = .026) but not in men ($p$ = .369). It is worth noting that the interaction term, financial inclusion × social network, did not achieve statistical significance β = -.358, $p$ = .984) indicating that the effect of financial inclusion on ADL impairment did not differ between those with

**Table 3. Multivariate OLS regressions predicting IADL impairment with composite score of financial inclusion and covariates.**

| | Model 1 –IADL: Overall | | Model 2 –IADL: women | | Model 3 –IADL: men | | Model 4 –IADL: Interaction– financial inclusion × social network | |
|---|---|---|---|---|---|---|---|---|
| | β | (SE) | β | (SE) | β | (SE) | β | (SE) |
| Financial inclusion (0–8; higher values indicating higher levels of financial inclusion) | -0.359* | (0.169) | -0.445* | (0.199) | -0.316 | (.352) | -0.358* | (0.170) |
| Age (years, ref: 50–59) | | | | | | | | |
| 60–69 | 0.979*** | (0.184) | 0.839*** | (0.234) | 1.458*** | (0.332) | 0.980*** | (0.184) |
| 70+ | 1.527*** | (0.192) | 1.406*** | (0.244) | 1.953*** | (0.357) | 1.527*** | (0.192) |
| Gender (ref: Males) | | | | | | | | |
| Females | 0.041 | (0.164) | - | - | - | - | 0.041 | (0.164) |
| Residence (ref: rural) | | | | | | | | |
| Urban | -0.039 | (0.150) | -0.241 | (0.190) | 0.422 | (0.263) | -0.039 | (0.150) |
| Education (ref: Primary/ none) | | | | | | | | |
| Secondary | 0.325 | (0.277) | 0.515 | (0.406) | 0.365 | (0.405) | 0.325 | (0.277) |
| Higher | -0.041 | (0.328) | 0.084 | (0.558) | -0.033 | (0.440) | -0.041 | (0.328) |
| Employment (ref: unemployed) | | | | | | | | |
| Employed | -0.421** | (0.159) | -0.203 | (0.206) | -0.932*** | (0.277) | -0.421** | (0.159) |
| Living alone | 0.020 | (0.167) | 0.058 | (0.217) | -0.029 | (0.293) | 0.020 | (0.167) |
| Social networks score | -0.271 | (0.149) | -0.142 | (0.191) | -0.625* | (0.260) | -0.272 | (0.161) |
| Self-reported health (ref: poor) | | | | | | | | |
| Fair | -2.777*** | (0.281) | -2.898*** | (0.364) | -2.563*** | (0.459) | -2.777*** | (0.281) |
| Good | -1.765*** | (0.247) | -2.055*** | (0.305) | -1.313** | (0.430) | -1.765*** | (0.247) |
| Very good | -1.006*** | (0.249) | -1.124*** | (0.308) | -1.048* | (0.439) | -1.007*** | (0.249) |
| Diagnosed chronic conditions | 0.862*** | (0.148) | 0.822*** | (0.186) | 0.983*** | (0.265) | 0.862*** | (0.148) |
| Psychological distress sore | 0.758*** | (0.156) | 0.512** | (0.192) | 1.392*** | (0.294) | 0.758*** | (0.156) |
| **Interaction effect:** financial inclusion × social network | | | | | | | -0.006 | (0.319) |
| Constant | 0.533*** | (0.154) | 0.701** | (0.250) | 0.503* | (0.239) | 0.533*** | (0.154) |
| Model diagnostics (-2 log likelihood) | 1162.375 | | 741.497 | | 397.442 | | 1162.375 | |
| Adjusted $R^2$ | 0.439 | | 0.406 | | 0.512 | | 0.439 | |

Beta-Coefficients are reported; Cluster-robust standard errors in parentheses.

Model 1: overall sample, Model 2: women; Model 3: men, Model 4: interaction analysis (financial inclusion × social networks).

***$p < 0.001$

**$p < 0.005$

*$p < 0.05$.

and without social networks. Additional analysis, showed that ownership of a bank account, withdrawing money from personal account and use of ATM were negatively associated with both ADL and IADL impairments. Further, whilst having credit union and mobile money accounts inversely related to ADL decline, having active health insurance was associated with a decrease in IADL limitations (Tables 4 and 5).

**Table 4. Multivariate OLS regressions predicting ADL impairment with specific financial service instrument.**

| Variable | 1 | | 2 | | 3 | | 4 | | 5 | | 6 | | 7 | | 8 | |
|---|---|---|---|---|---|---|---|---|---|---|---|---|---|---|---|---|
| | β | (SE) | β | (SE) | β | (SE) | β | (SE) | β | (SE) | β | (SE) | β | (SE) | β | (SE) |
| Withdrawal of money from a bank | -0.166* | (0.142) | | | | | | | | | | | | | | |
| Ownership of current/savings account | | | -0.335*** | (0.155) | | | | | | | | | | | | |
| Use of automatic teller machine card | | | | | -0.676* | (0.274) | | | | | | | | | | |
| Membership of a credit union | | | | | | | -0.153* | (0.198) | | | | | | | | |
| Ownership of "susu" union/ account | | | | | | | | | -0.107 | (0.161) | | | | | | |
| Access to loan from financial institution | | | | | | | | | | | 0.261 | (0.200) | | | | |
| Ownership of money from Mobile Money account | | | | | | | | | | | | | -0.054*** | (0.140) | | |
| Having active NHIS card | | | | | | | | | | | | | | | -0.481 | (0.273) |
| Potential confounders | √ | | √ | | √ | | √ | | √ | | √ | | √ | | √ | |
| Constant | -2.262***(.446) | | -2.277***(.443) | | -2.239***(.443) | | -2.337***(.441) | | -2.345***(.441) | | -2.348*** (.442) | | -2.304***(.452) | | -2.528***(.455) | |
| Adjusted $R^2$ | 0.303 | | 0.306 | | 0.307 | | 0.302 | | 0.302 | | 0.303 | | 0.302 | | 0.304 | |

Beta-Coefficients are reported; Cluster-robust standard errors in parentheses.

All Models were adjusted for theoretically relevant confounders: age, gender, rural/urban residence, educational level, employment status, living arrangement, social networks, self-rated health, psychological distress, chronic conditions.

*$p < .05$.

**$p < .005$.

***$p < .001$.

## Discussion

Access to and use of financial services have become an important policy issue in low- and middle-income countries where, notably, a large proportion of the population lacks access to basic financial services [12]. Based on a representative sample, this study investigated the association between financial inclusion and functional impairment of the fast-growing and vulnerable target population of community-dwelling older adults. Adjusting for potential confounders, OLS regressions showed that financial inclusion was independently associated with decreases in ADL and IADL indicators of physical health functioning. In addition, social networks and gender moderated the negative relationship between financial inclusion and functional impairment as financially included men and those with a stronger constellation of social networks experience even lower levels of functional impairment with particular emphasis to ADL limitation compared to their respective counterparts. To our knowledge, this is the first study to provide estimates of the interactive effects of financial inclusion, social networks and gender on physical health functioning for older people in a low- and middle-income context with multiple indicators and techniques with adjustments for theoretically potential covariates.

**Table 5. Multivariate OLS regressions predicting IADL impairment with specific financial service instrument.**

| Variable | 1 | | 2 | | 3 | | 4 | | 5 | | 6 | | 7 | | 8 | |
|---|---|---|---|---|---|---|---|---|---|---|---|---|---|---|---|---|
| | β | (SE) | β | (SE) | β | (SE) | β | (SE) | β | (SE) | β | (SE) | β | (SE) | β | (SE) |
| Withdrawal of money from a bank | -0.246* | (0.155) | | | | | | | | | | | | | | |
| Ownership of current/savings account | | | -0.071** | (0.166) | | | | | | | | | | | | |
| Use of automatic teller machine card | | | | | -0.179* | (0.298) | | | | | | | | | | |
| Membership of a credit union | | | | | | | -0.165 | (0.213) | | | | | | | | |
| Ownership of "susu" union/account | | | | | | | | | 0.030 | (0.172) | | | | | | |
| Loan from financial institution | | | | | | | | | | | 0.105 | (0.211) | | | | |
| Ownership of money from Mobile Money account | | | | | | | | | | | | | 0.037 | (0.154) | | |
| Having active NHIS card | | | | | | | | | | | | | | | -0.233** | (0.268) |
| Potential confounders | √ | | √ | | √ | | √ | | √ | | √ | | √ | | √ | |
| Constant | -3.609***(.510) | | -3.706***(.507) | | -3.690***(.508) | | -3.725***(.505) | | -3.723*** (.506) | | -3.725*** (.506) | | -3.751*** (.518) | | -3.642***(.514) | |
| Adjusted $R^2$ | 0.440 | | 0.438 | | 0.438 | | 0.439 | | 0.438 | | 0.438 | | 0.438 | | 0.439 | |

Beta-Coefficients are reported; Cluster-robust standard errors in parentheses.

All Models were adjusted for theoretically relevant confounders: age, gender, rural/urban residence, educational level, employment status, living arrangement, social networks, self-rated health, psychological distress, chronic conditions.

*$p < .05$.

**$p < .005$.

***$p < .001$.

Previous findings, largely, from advanced countries have shown that participation in the financial market can favorably influence individual mental health [14, 15] and that the context of financial inclusion might particularly provide high levels of benefit especially among the vulnerable population groups [7, 29, 30]. Others have also estimated the role of financial inclusion in overall health, health services use and well-being of populations [2, 18, 20, 26]. Our results add important insight into the growing streams of research and complement previous studies on the association between financial inclusion and well-being by suggesting that financial inclusion can offer older individuals better functional health. Access to financial services and financial market participation may be associated with lower levels of functional impairment through several fundamental mechanisms or hypotheses. First, studies have identified low level of physical activity as a major modifiable risk factor for functional impairments [31, 41, 42]. The ability to remain active and engaged in the financial market and its related day-to-day activities may directly contribute to older people's functional strength and capacities [2]. This may counteract or delay activity limitations associated with older age [28]. Most importantly, financial capability underlines the importance of individuals acquiring knowledge, skills and desired services, and managing their finances that can build financial security [43]. Research demonstrates that access to financial services such as ownership of bank account and health insurance are directly associated with improved financial stability and reduced mental disorders such as stress, anxiety, depression and worries [15, 44]. Thus,

improved mental health and psychological wellbeing have strongly related to better physical health and functioning [4, 45]. Our findings, therefore, support previous evidence from developed economies that increasing financial sector participation is a relevant pathway toward improved physical health and independence among older people. However, our findings were generally inconsistent with the only study we could find that did not establish any association between being banked and physical health among older Hispanics [15]. We may relate this inconsistency to the socio-cultural differences between the US and African country older populations.

Findings demonstrate that financial inclusion is an important and gendered factor shaping the magnitude of physical health in older age. Both financially included women and men showed associations with decreased ADL impairments but the rate of decrease was stronger among men (66%) compared to women (44%). The implication is that high levels of financial inclusion protected men against ADL limitations than women. In terms of IADL, the relationship between financial inclusion and physical health did not reach significance. These observations may be akin to the hypothesis that the involvement in the financial market and the way financial inclusion responses to health status are different for men and women [10, 29]. These findings suggest that improvement in access to financial services may be important policy intervention for especially men in reducing functional impairment.

The finding that social networks significantly moderated the link between financial inclusion and ADL limitations is worthy of note. Our findings indicate that stronger social relationships in later life may reinforce the negative association between financial inclusion and activity limitations. Research has shown that generally, social networks and social support represent a key coping resource for stressors related to disability and functional impairments [46, 47]. This finding is also related to the notion that socially connected individuals may have a greater tendency to become financially included [10, 31]. The implications of this finding is, that policy makers need to be aware of the key roles of social networks and interpersonal relationships when considering financial inclusion as a tool to mitigating functional impairments among older people in sub-Saharan Africa.

The results of this study should be considered within its limitations. The sample was limited to community-dwelling adults aged 50 years and over. This could limit generalizability to younger individuals from other geographic regions. However, the findings may provide some important baseline data and key insights on how financial inclusion affects functional health in older age and for future research endeavors. The assessment of financial inclusion, ADL and IADL limitations was self-reported, introducing scope for social and subjective bias although self-report may be one of the best and certainly most convenient way to capture social views [48]. Moreover, the cross-sectional design adopted in this study may defy causal and directional conclusions. Future research exploring the relationships between financial inclusion and functional health would benefit from longitudinal data. Also, in-depth qualitative exploration of this relationship may increase understanding and inform public health and policy directions. Despite the limitations, our study takes an important step toward developing a more nuanced understanding of the potential role of financial inclusion in functional status in later life, a neglected relationship in sub-Saharan Africa. The main strengths of this study are the large representative sample and the use of a validated, reliable and internationally recognized questionnaire to measure functional limitations [37, 38] in an innovative setting. Our findings are valuable for both academic and policy-related purposes, given the growing need for older populations to maintain independence toward successful aging. This is particularly important in resource-poor settings where access to and use of financial services are generally infant.

## Conclusions

The effect of financial inclusion on functional status at older ages remains a relatively under-explored subject in sub-Saharan Africa. The multivariate analyses revealed a significant association between financial inclusion and physical health functioning in Ghanaian older adults although we found no association for men in terms of IADL. Most importantly, financial inclusion and social networks significantly interacted to further decrease ADL impairment such that highly connected individuals were less likely to suffer from ADL decline following financial inclusion. This supports the view that older sub-Saharan African adults should be encouraged and empowered to be financially included. Effective strategies to maintain functional independence in this population group would be through targeted intervention programs to ensure easy access and use of financial services in the context of improved social and interpersonal networks.

## Supporting information

**S1 Data.**
(SAV)

## Author Contributions

**Conceptualization:** Razak M. Gyasi, Siaw Frimpong, Gilbert Kwabena Amoako, Anokye M. Adam.

**Data curation:** Razak M. Gyasi, Siaw Frimpong, Gilbert Kwabena Amoako, Anokye M. Adam.

**Formal analysis:** Razak M. Gyasi, Siaw Frimpong, Anokye M. Adam.

**Funding acquisition:** Razak M. Gyasi.

**Investigation:** Razak M. Gyasi.

**Methodology:** Razak M. Gyasi, Gilbert Kwabena Amoako, Anokye M. Adam.

**Project administration:** Razak M. Gyasi.

**Resources:** Razak M. Gyasi.

**Software:** Razak M. Gyasi, Anokye M. Adam.

**Supervision:** Razak M. Gyasi, Gilbert Kwabena Amoako.

**Validation:** Razak M. Gyasi.

**Visualization:** Razak M. Gyasi.

**Writing – original draft:** Razak M. Gyasi, Siaw Frimpong, Gilbert Kwabena Amoako, Anokye M. Adam.

**Writing – review & editing:** Razak M. Gyasi, Siaw Frimpong, Gilbert Kwabena Amoako, Anokye M. Adam.

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
