## [Decision Letter · Decision Letter 0]

26 Nov 2020

PONE-D-20-09674

Access to financial services and physical health functioning in later life: Results of a population-based study in older age

PLOS ONE

Dear Dr. Gyasi,

Thank you for submitting your manuscript to PLOS ONE. After careful consideration, we feel that it has merit but does not fully meet PLOS ONE’s publication criteria as it currently stands. Therefore, we invite you to submit a revised version of the manuscript that addresses the points raised during the review process.

The manuscript has been evaluated by two reviewers, and their comments are available below.

The reviewers have raised a number of concerns that need attention. They request additional information on the rationale and theoretical framework applied, and on some methodological aspects of the study; their comments should also be considered in a revised Discussion to fully illustrate the limitations of the study. 

Could you please revise the manuscript to carefully address the concerns raised?

We look forward to receiving your revised manuscript.

Kind regards,

Carmen Melatti

Associate Editor

PLOS ONE

Journal Requirements:

3. Please include additional information regarding the survey or questionnaire used in the study and ensure that you have provided sufficient details that others could replicate the analyses. For instance, if you developed a questionnaire as part of this study and it is not under a copyright more restrictive than CC-BY, please include a copy, in both the original language and English, as Supporting Information. Moreover, please include more details on how the questionnaire was pre-tested, and whether it was validated.

Reviewers' comments:

Reviewer's Responses to Questions

**Comments to the Author**

1. Is the manuscript technically sound, and do the data support the conclusions?

Reviewer #1: Partly

Reviewer #2: Partly

2. Has the statistical analysis been performed appropriately and rigorously? 

Reviewer #1: No

Reviewer #2: No

3. Have the authors made all data underlying the findings in their manuscript fully available?

Reviewer #1: No

Reviewer #2: Yes

4. Is the manuscript presented in an intelligible fashion and written in standard English?

Reviewer #1: No

Reviewer #2: Yes

5. Review Comments to the Author

Reviewer #1: The paper tries to answer whether financial inclusion is associated with older-age physical health functioning and if the association differs by social relationships. There are several issues which are not accurate or should be correct before it will be possible to deal and ask whether the paper is a propriate one.

1. The first and the most bothered issues concerns the theoretical framework. Even though there is a short paragraph of literature background, it is not clear what is the theory behind this paper and why do the authors assume that financial inclusion is associated with older-age physical health functioning. Why is that? what is the rational behind this assumption?

2. There is a need to add more specific hypothesis regarding potential differences between economic classes.

3. What is the rational for classifying some of the variables as control variables? some as moderating variables?

4. The authors should add both a theoretical model as well as an empirical model and to explain it.

5. What was the specific criteria for including variables in the models?

6. Why did you use interactions? what is the added value of using it? I think that using interactions has a very problematic and very minor effect here.

7. There is a need to re-estimate the models so that there will be several steps - a first step without the financial inclusion variable and a second stage that contains it.

8. The discussion present very poorly the authors way of analyzing the results. There is a need to re-write this chapter and to add a very detailed explanation - using the authors own words - about the results and there meanings.

9. In addition with all of the above - there is a need to re-write the abstract. Some of the the paragraphs in it are vague.

Reviewer #2: The author investigates the association between financial inclusion and physical limitations (measured as ADLs and IADLs) among older individuals in Ghana.

The research question is interesting and of particular relevance for a developing country.

The literature review is clearly written and exhaustive.

However, there are some important weaknesses in the empirical section:

- the specified model reports among the controls measures of health (self-reported health and number of chronic disease). These are alternative outcome variables and should not appear on the right-hand side of the estimated equation. These additional controls make it difficult to interpret the results provided.

- as highlighted by the author in the limitation section it is hard to give a causal interpretation to the estimates. There are unobserved characteristics that are likely to affect both financial inclusion and health and one would need repeated observations for the same individual to get rid of this selection problem. In fact, we might expect that more financially prepared individuals are also those in better health, and the significant parameter for financial inclusion might simply reflect this.

- About the role of social networks, according to the estimates the beneficial effect of financial inclusion on physical health is stronger for more socially active individuals. However, being more socially connected is positively associated with ADL and IADL. This might reflect the fact that physically impaired individuals are more in need of help and the social network score captures the presence of caregivers.

- it is not clear to me whether the survey includes additional information that can be used to mitigate the identification problems listed above. For sure, it might be interesting to analyze the single items included in the financial inclusion measure. At least providing descriptive statistics and commenting on their relevance in the institutional context considered.

Minor issues:

- in the third paragraph of the introduction the author states that “Research, particularly in advanced countries shows that financial literacy/inclusion are positively associated with common mental health problems”, but it is the opposite: financial inclusion is negatively associated with mental health problems.

- Table 1 reporting descriptive statistics is confusing. For continuous variables mean and sd are reported, whereas for binary variables the table shows the frequencies. I suggest revising this table.

6. PLOS authors have the option to publish the peer review history of their article (what does this mean?). If published, this will include your full peer review and any attached files.

Reviewer #1: No

Reviewer #2: No

---

## [Author Response · Author response to Decision Letter 0]

7 Feb 2021

RESPONSES TO COMMENTS

Reviewer #1: 

The paper tries to answer whether financial inclusion is associated with older-age physical health functioning and if the association differs by social relationships. There are several issues which are not accurate or should be correct before it will be possible to deal and ask whether the paper is appropriate one.

Response: We thank the reviewer for this observation. Based on the comments below, we have worked on the entire manuscript and attempted to respond to the various issues raised. This opportunity and guidance has been very valuable in improving the paper overall. 

1. The first and the most bothered issues concerns the theoretical framework. Even though there is a short paragraph of literature background, it is not clear what is the theory behind this paper and why do the authors assume that financial inclusion is associated with older-age physical health functioning. Why is that? What is the rationale behind this assumption?

Response: We note carefully these comments on theoretical foundation of our paper. We appreciate the specific role of relevant theories and theoretical development in social research and our study. In this regard we have expanded the discussion of the relevant theoretical explanations including financial capability. We have espoused that access to financial services such as health insurance for the uninsured and access to and ownership of bank accounts improves the wellbeing by providing financial protection, reducing stress, and ultimately, enhance both mental and physical health outcomes of older people who may lack regularly incomes due to retirement and other socioeconomic circumstances. That is to say, that improving older adults’ ability to manage their finances remains a fundamental step to promoting financial capabilities, and in turn, improves their wellbeing and overall health [19, 20]. Pages 4 of the revised version of the draft incorporate these theoretical foundations. 

2. There is a need to add more specific hypothesis regarding potential differences between economic classes.

Response: We fully agree with our reviewer for the suggestion to included specific hypothesis on the potential differences between economic classes of the study sample. Given the specific relationships we observed in the literature review, we did not aim to evaluate the direct relationship between different economic groups of our respondents and their functional or physical health. We maintained that being financially included is a proxy of economic condition of older persons and could be very important in later life even than having income. It is also important to note that older persons might have converted their incomes into various economic values such as bonds or insurance which had made them financially included. We believe that estimating the direct relationship between income and health outcomes although an important issue, did not fall within the remit of our study. Further research might consider this. 

3. What is the rational for classifying some of the variables as control variables? Some as moderating variables?

Response: Variables selected as the control variables were based on the observations of the previous studies, particularly as regards their relationships with functional health and their ability to influence the latter.

4. The authors should add both a theoretical model as well as an empirical model and to explain it.

Response: This has been done under the introduction. 

5. What was the specific criteria for including variables in the models?

Response: As we have responded query point 7 in detail below, we initially selected the control variables in the models and then added the prime explanatory variable. However, we chose to maintain and to work with the Full Model since our interest was rather to interpret the adjusted results of the association of financial inclusion with physical health functioning. 

6. Why did you use interactions? What is the added value of using it? I think that using interactions has a very problematic and very minor effect here.

Response: The variables included in the interaction analysis and all the analyses we performed were motivated in the introduction of the paper and also provided a hypothesis for each based on the evidence in the literature. Based on prior evidence, we examined social networks and support thereof as possible moderator of the association between financial inclusion and physical health. We, therefore, did not fished for specific results we were interested in. The moderation analysis in this case provided a significant value and offered very important explanation of the association between financial inclusion and physical health of older people. We have highlighted each of these in the background of the paper. 

7. There is a need to re-estimate the models so that there will be several steps - a first step without the financial inclusion variable and a second stage that contains it.

Response: Whilst we fully agree with the reviewer, we thinks this is just a repetition of effort especially when our interest rests upon the final adjusted model. We did not make any assumptions for the estimation of the unadjusted models. We, therefore, resolved to maintain the original shape of our analysis without taking a cognizance of the unadjusted results. Thank you. 

8. The discussion present very poorly the authors’ way of analyzing the results. There is a need to re-write this chapter and to add a very detailed explanation - using the authors own words - about the results and their meanings.

Response: Thank you for the suggestion. We have restructured and improved portions of the discussion section by providing detailed highlights for our specific and innovative results.

9. In addition with all of the above - there is a need to re-write the abstract. Some of the paragraphs in it are vague.

Response: This is an important observation. We humbly wish to acknowledge that the reviewer is correct to state that we reconsider the write-up of the abstract. We have worked on the entire abstract and have improved upon it.

Reviewer #2: 

The author investigates the association between financial inclusion and physical limitations (measured as ADLs and IADLs) among older individuals in Ghana. The research question is interesting and of particular relevance for a developing country. The literature review is clearly written and exhaustive.

Response: We are grateful to this reviewer for this positive comment on the relevance and how interesting our paper looks. We especially note the recommendation. 

However, there are some important weaknesses in the empirical section:

- The specified model reports among the controls measures of health (self-reported health and number of chronic disease). These are alternative outcome variables and should not appear on the right-hand side of the estimated equation. These additional controls make it difficult to interpret the results provided.

Response: Thanks for the observation. All control variables included in these analyses were selected based on prior evidence in the literature. Aside from our prime exposure variable, other physical health-related factors have been reported to relate somewhat with functional health (ADL and IADL) including self-rated health and chronic conditions. In this regard, we explored the relationship by adjusting for any potential variable that may influence functional impairment. Whist we appreciate that both physical impairment, self-rated health and chronic diseases are considered as elements of physical health, they are not the same and also measured differently. Controlling for the former will yield a better estimates for the relationship between financial inclusion and ADL and IADL. 

- As highlighted by the author in the limitation section it is hard to give a causal interpretation to the estimates. There are unobserved characteristics that are likely to affect both financial inclusion and health and one would need repeated observations for the same individual to get rid of this selection problem. In fact, we might expect that more financially prepared individuals are also those in better health, and the significant parameter for financial inclusion might simply reflect this.

Response: This is indeed a critical observation and we could not have agreed much with the reviewer. Our data, being cross-sectional in nature, were limited in terms of its ability to estimate and explore the cause and directional relationships between variables. This is a very important limitation as we acknowledged. However, these findings will serve as a baseline to guide any future studies that will explore the associations between financial inclusion and health outcomes particularly in developing countries. We have proposed that future research takes account of a longitudinal data so as to be able to deal with this challenge. 

- About the role of social networks, according to the estimates the beneficial effect of financial inclusion on physical health is stronger for more socially active individuals. However, being more socially connected is positively associated with ADL and IADL. This might reflect the fact that physically impaired individuals are more in need of help and the social network score captures the presence of caregivers.

Response: We appreciate this important comment. We have had a second look at the estimations reported in relation to social networks and ADL and IADL impairments. Unfortunately, we provided the reverse of the association in the write-up by considering higher levels of social support as reference. Upon re-estimation of the results, we found a rather negative association between social support and both ADL and IADL. We have there rectify this oversight in the revised version of the manuscript. 

- It is not clear to me whether the survey includes additional information that can be used to mitigate the identification problems listed above. For sure, it might be interesting to analyze the single items included in the financial inclusion measure. At least providing descriptive statistics and commenting on their relevance in the institutional context considered.

Response: We have provided additional analyses and estimated the impact of each of the financial inclusion instrument on both ADL and IADL as presented in Tables 4 and 5 respectively. 

Minor issues:

- In the third paragraph of the introduction the author states that “Research, particularly in advanced countries shows that financial literacy/inclusion are positively associated with common mental health problems”, but it is the opposite: financial inclusion is negatively associated with mental health problems.

Response: We appreciate this critical observation and we fully agree with you. This was an oversight. We have, therefore, rectified the error. 

- Table 1 reporting descriptive statistics is confusing. For continuous variables mean and SD are reported, whereas for binary variables the table shows the frequencies. I suggest revising this table.

Response: We have rectified this anomaly. Thank you. 

Thank you again for the opportunity to resubmit our paper to PloS ONE. 

Yours sincerely,

(Corresponding author)

---

## [Decision Letter · Decision Letter 1]

6 Apr 2021

PONE-D-20-09674R1

How does financial health affect functional impairment in old age? Exploring social networks and gender roles

PLOS ONE

Dear Dr. Gyasi

Thank you for submitting your manuscript to PLOS ONE. After careful consideration, we feel that it has merit but does not fully meet PLOS ONE’s publication criteria as it currently stands. Therefore, we invite you to submit a revised version of the manuscript that addresses the points raised during the review process.

ACADEMIC EDITOR:

One of the reviewer claims that you mixed between references from developed and underdeveloped countries, which delivers incorrect information in many places. In order to progress with the evaluation process the authors have to focus and emphasis that the paper is related to a relevant population.It might also be of important to compare the results with the situation with developed countries and conclude with suggesting forward steps that developed countries should maintain.

We look forward to receiving your revised manuscript.

Kind regards,

Aviad Tur-Sinai, Ph.D

Academic Editor

PLOS ONE

Reviewers' comments:

Reviewer's Responses to Questions

**Comments to the Author**

1. If the authors have adequately addressed your comments raised in a previous round of review and you feel that this manuscript is now acceptable for publication, you may indicate that here to bypass the “Comments to the Author” section, enter your conflict of interest statement in the “Confidential to Editor” section, and submit your "Accept" recommendation.

Reviewer #3: All comments have been addressed

Reviewer #4: (No Response)

2. Is the manuscript technically sound, and do the data support the conclusions?

Reviewer #3: Partly

Reviewer #4: (No Response)

3. Has the statistical analysis been performed appropriately and rigorously? 

Reviewer #3: Yes

Reviewer #4: (No Response)

4. Have the authors made all data underlying the findings in their manuscript fully available?

Reviewer #3: Yes

Reviewer #4: (No Response)

5. Is the manuscript presented in an intelligible fashion and written in standard English?

Reviewer #3: Yes

Reviewer #4: (No Response)

6. Review Comments to the Author

Reviewer #3: The revised manuscript presents an overall good response to points raised in the original version.

I would like to state that the revised manuscript should discuss the well-known phenomena named "social determinants of health". the paper deals with economic literacy and other considerations while this is not unique.

however, i feel that the presented work adds information to this important issue and should thus be accepted for publication.

Reviewer #4: The study deals with the relationships between financial inclusion and functional health among older adults, and in relation to gender. Although the issue is important, it has been studied extensively in developed countries for many years. Even the conclusion is well known "Findings show that a higher degree of financial inclusion in the context of social networks has a beneficial effect on functional health in later life ". Therefore the study might be of important in relation to developing countries, where this issue is not recognized enough. Thus, the fact that the study is related to the sub-Saharan Africa should be emphasized in the title and along the paper, from the first sentence of the abstract, up to the discussion.

For example, the first sentence in the abstract starts with: It remains poorly understood if financial inclusion is associated with older [1] age physical health functioning and if the association differs by social relationships", is not true! It might be true for developing countries such as sub-Saharan Africa. This point should be corrected all over the manuscript.

In relation to this point, the authors mixed between references from developed and underdeveloped countries, which makes mish mash among the references, which delivers incorrect information in many places. It is sticking out in the introduction, and less dominant in the discussion, where the authors presented comparisons to less affluent population in western countries.

I believe that before continuing the evaluation of this study, the authors have to focus and emphasis that the paper is related to a relevant population. It might also be of important to compare their results with the situation with developed countries and conclude with the forward steps that developed countries should maintain.

Consequently, , I think that the paper could be evaluated for publication, only after the authors make the appropriate corrections.

7. PLOS authors have the option to publish the peer review history of their article (what does this mean?). If published, this will include your full peer review and any attached files.

Reviewer #3: No

Reviewer #4: No

---

## [Author Response · Author response to Decision Letter 1]

22 Apr 2021

RESPONSES TO COMMENTS

Reviewer #3: 

The revised manuscript presents an overall good response to points raised in the original version.

Response: Thanks to the reviewer for the observation regarding the improvement in our manuscript upon initial revision. We appreciate your initial comments and suggestions. 

I would like to state that the revised manuscript should discuss the well-known phenomena named "social determinants of health". The paper deals with economic literacy and other considerations while this is not unique, I feel that the presented work adds information to this important issue and should thus be accepted for publication. 

Response: We note carefully these comments on the highlight of social determinants of health. We have provided a brief explanation of the concept in our paper. Thank you for the suggestion. 

Reviewer #4: 

The study deals with the relationships between financial inclusion and functional health among older adults, and in relation to gender. Although the issue is important, it has been studied extensively in developed countries for many years. Even the conclusion is well known "Findings show that a higher degree of financial inclusion in the context of social networks has a beneficial effect on functional health in later life". Therefore the study might be of important in relation to developing countries, where this issue is not recognized enough. Thus, the fact that the study is related to the sub-Saharan Africa should be emphasized in the title and along the paper, from the first sentence of the abstract, up to the discussion.

Response: We are grateful to this reviewer for this positive comment on the relevance and how interesting our paper looks. We especially note the recommendation. We could not have agreed with the reviewer much on the assertion that this topic has extensively evaluated in advanced countries but less evidence exists in developing countries including sub-Saharan Africa. Throughout the manuscript, we have related our arguments to the sub-Saharan African settings. 

For example, the first sentence in the abstract starts with: It remains poorly understood if financial inclusion is associated with older [1] age physical health functioning and if the association differs by social relationships", is not true! It might be true for developing countries such as sub-Saharan Africa. This point should be corrected all over the manuscript.

Response: Related to the above response, we have edited and rectified the issues and strongly related our write up and arguments to low- and middle-income countries including sub-Saharan Africa.

In relation to this point, the authors mixed between references from developed and underdeveloped countries, which makes mishmash among the references, which delivers incorrect information in many places. It is sticking out in the introduction, and less dominant in the discussion, where the authors presented comparisons to less affluent population in western countries.

Response: This is rectified. 

I believe that before continuing the evaluation of this study, the authors have to focus and emphasis that the paper is related to a relevant population. It might also be of importance to compare their results with the situation with developed countries and conclude with the forward steps that developed countries should maintain.

Response: This has been done in the revised manuscript. 

Consequently, I think that the paper could be evaluated for publication, only after the authors make the appropriate corrections.

Response: This is noted in good faith, thank you. 

Thank you again for the opportunity to resubmit our paper to PLOS ONE. 

Yours sincerely,

(Corresponding author)

---

## [Decision Letter · Decision Letter 2]

10 May 2021

How does financial inclusion affect functional impairment among aging adults in Ghana? Exploring social networks and gender roles

PONE-D-20-09674R2

Dear Dr. Gyasi,

We’re pleased to inform you that your manuscript has been judged scientifically suitable for publication and will be formally accepted for publication once it meets all outstanding technical requirements.

Kind regards,

Prof. Aviad Tur-Sinai, Ph.D

Guest Editor

PLOS ONE

Additional Editor Comments (optional):

Reviewers' comments:

Reviewer's Responses to Questions

**Comments to the Author**

1. If the authors have adequately addressed your comments raised in a previous round of review and you feel that this manuscript is now acceptable for publication, you may indicate that here to bypass the “Comments to the Author” section, enter your conflict of interest statement in the “Confidential to Editor” section, and submit your "Accept" recommendation.

Reviewer #3: All comments have been addressed

Reviewer #4: (No Response)

2. Is the manuscript technically sound, and do the data support the conclusions?

Reviewer #3: Yes

Reviewer #4: (No Response)

3. Has the statistical analysis been performed appropriately and rigorously? 

Reviewer #3: Yes

Reviewer #4: (No Response)

4. Have the authors made all data underlying the findings in their manuscript fully available?

Reviewer #3: Yes

Reviewer #4: (No Response)

5. Is the manuscript presented in an intelligible fashion and written in standard English?

Reviewer #3: Yes

Reviewer #4: (No Response)

6. Review Comments to the Author

Reviewer #3: The revised manuscript is suitable for publication in PLOS as all comments have been addressed. good luck

Reviewer #4: (No Response)

7. PLOS authors have the option to publish the peer review history of their article (what does this mean?). If published, this will include your full peer review and any attached files.

Reviewer #3: No

Reviewer #4: No

---

## [Editor Report · Acceptance letter]

21 May 2021

PONE-D-20-09674R2 

Financial inclusion and physical health functioning among aging adults in the sub-Saharan African context: Exploring social networks and gender roles 

Dear Dr. Gyasi:

I'm pleased to inform you that your manuscript has been deemed suitable for publication in PLOS ONE. Congratulations! Your manuscript is now with our production department. 

Kind regards, 

on behalf of

Prof. Aviad Tur-Sinai 

Guest Editor

PLOS ONE